# Identification of SIBO Subtypes along with Nutritional Status and Diet as Key Elements of SIBO Therapy

**DOI:** 10.3390/ijms25137341

**Published:** 2024-07-04

**Authors:** Justyna Paulina Wielgosz-Grochowska, Nicole Domanski, Małgorzata Ewa Drywień

**Affiliations:** 1Department of Human Nutrition, Institute of Human Nutrition Sciences, Warsaw University of Life Sciences, 02-776 Warsaw, Poland; justyna_wielgosz@sggw.edu.pl; 2Faculty of Pharmaceutical Sciences, University of British Columbia, Vancouver, BC V6T 1Z3, Canada; nicole.domanski@ubc.ca

**Keywords:** SIBO, dysbiosis, microbiota, treatment, diet, fiber, deficiency, breath test

## Abstract

Small intestinal bacterial overgrowth (SIBO) is a pathology of the small intestine and may predispose individuals to various nutritional deficiencies. Little is known about whether specific subtypes of SIBO, such as the hydrogen-dominant (H+), methane-dominant (M+), or hydrogen/methane–dominant (H+/M+), impact nutritional status and dietary intake in SIBO patients. The aim of this study was to investigate possible correlations between biochemical parameters, dietary nutrient intake, and distinct SIBO subtypes. This observational study included 67 patients who were newly diagnosed with SIBO. Biochemical parameters and diet were studied utilizing laboratory tests and food records, respectively. The H+/M+ group was associated with low serum vitamin D (*p* < 0.001), low serum ferritin (*p* = 0.001) and low fiber intake (*p* = 0.001). The M+ group was correlated with high serum folic acid (*p* = 0.002) and low intakes of fiber (*p* = 0.001) and lactose (*p* = 0.002). The H+ group was associated with low lactose intake (*p* = 0.027). These results suggest that the subtype of SIBO may have varying effects on dietary intake, leading to a range of biochemical deficiencies. Conversely, specific dietary patterns may predispose one to the development of a SIBO subtype. The assessment of nutritional status and diet, along with the diagnosis of SIBO subtypes, are believed to be key components of SIBO therapy.

## 1. Introduction

The gut microbiota represents the second gene pool in the human body and participates in the regulation of various biological processes [1]. The mutually symbiotic relationship between microorganisms and the human body is vital for sustaining our health. However, a lack of effective communication between the host organism and the microbiome can contribute to the development of human diseases [2]. The gut microbiota serves as an incredible reservoir of microorganisms. Variations in the composition of the gut microbiota occur depending on the particular anatomical area being examined [3]. These differences are chiefly influenced by factors like pH level and concentrations of oxygen. The greatest bacterial colonization occurs predominantly in the large intestine [4]. The small intestine, under normal physiological conditions, is colonized by a specific number of microorganisms, from 10^4–5^ CFU/mL in the proximal region to 10^7–8^ CFU/mL in the distal part [5]. However, when the host’s defence mechanisms are compromised, bacterial translocation and overgrowth can occur in the small or large intestine [6,7,8].

Small intestinal bacterial overgrowth (SIBO) and intestinal methanogen overgrowth (IMO) are two types of gut microbiota disruption [9]. The overgrowth of bacteria in SIBO and IMO can be associated with the migration of bacteria from the upper aerodigestive tract or from the colon [10]. If the overgrowth is predominately from oral microbiota, it is often characterized by bacteria such as *Prevotella* and *Streptococcus gramineus*, whereas the bacteria involved in the pathogenesis of coliform SIBO are more likely to include *Klebsiella pneumoniae*, *Escherichia coli*, *Clostridioides* sp., *Proteus mirabilis* and *Enterococcus* sp. [10,11]. Despite extensive research on SIBO in recent years, it remains an overlooked issue for many specialists. However, thanks to the latest non-invasive diagnostics methods, such as SIBO breath tests, diagnosis is becoming more common [6]. Breath tests enable the measurement of gases, such as hydrogen (H_2_) and methane (CH_4_), produced by intestinal bacteria as a result of carbohydrate fermentation and facilitate the identification of SIBO subtypes such as hydrogen-dominant, methane-dominant or hydrogen/methane-dominant [12,13]. In most cases of SIBO, H_2_ is the primary byproduct, and predominant H_2_ producers have been linked to more symptoms of diarrhea and higher levels of *Enterobacteriaceae domain* [14]. However, approximately 30% of individuals with SIBO harbor methane-producing Archaea such as *Methanobrevibacter smithii* in their gut [15]. Increased concentrations of exhaled methane have been correlated with more symptoms of constipation and an abundance of *Methanobrevibacter smithii*, as well as *Methanosphaera stadtmanae* and *Methanomassiliicoccus luminyensis* [14,16]. The primary cause of SIBO is identified as increased permeability to lipopolysaccharides (LPSs), which trigger an inflammatory response and lead to chronic inflammation [17]. However, it is important to remember that SIBO is just the tip of the iceberg—a dysfunction of many organs and a variety of diseases may contribute to excessive bacterial overgrowth and lead to SIBO [18]. Common gastrointestinal symptoms like abdominal pain, nausea, bloating, gas, diarrhea and/or constipation are considered the typical symptoms of dysbiosis, and consequently of SIBO as well [19]. Moreover, these issues can result in malabsorption, leading to nutritional deficiencies (vitamin A, D, E, B12), hypoproteinemia, anemia and weight loss [4,20,21,22]. While SIBO is not a life-threatening condition, it can worsen the patient’s health and underlying comorbidities [23,24,25,26].

Diet is a modifiable factor that plays a crucial role in shaping the composition, diversity and stabilityof the gut microbiota [27,28]. A diet rich in fiber and plant-based foods, supplemented with prebiotics and low in choline and fat is generally acknowledged to predispose one to a healthy microbiota [29]. Conversely, a regimen deficient in fiber, fermentable oligosaccharides, disaccharides, monosaccharides and polyols (FODMAPs), and characterized by an elevated intake of omega-6 fatty acids typical of a Western dietary pattern, may predispose one to dysbiotic conditions [29,30]. However, an approach in which proper care for gut microbiota nutrition is neglected may lead to nutritional deficiencies and microbiota alteration [31,32,33].

In treating SIBO, it is essential to focus on the proper functioning of the entire organism, including clinical symptoms and nutritional status, rather than solely attempting to eradicate or starve excessive microbial growth in the small intestine. Individuals experiencing gastrointestinal symptoms frequently turn to prolonged dietary restrictions, sometimes neglecting the significance of restoring microbial equilibrium in the gut [27,34]. Furthermore, patients blindly choose to consume only permitted products, often leading to an inadequate intake of fiber, calcium or iron due to poor meal planning [35,36,37]. However, it remains unclear whether a specific subtype of SIBO may predispose patients to a particular clinical presentation (i.e., nausea, diarrhea, constipation), which influences their dietary choices and patterns, or whether a patient’s baseline diet and nutritional intake may increase the risk of developing a specific SIBO subtype. The objective of this article is to investigate possible correlations between dietary intake, serum levels of biochemical parameters and SIBO subtypes in newly diagnosed SIBO patients.

## 2. Results

### 2.1. Demographic and Anthropometric Characteristics

In the study, there was a combined total of 67 participants who had recently received a diagnosis of SIBO and who were distributed among three distinct SIBO groups. Table 1 depicts demographic and anthropometric characteristics categorized by SIBO subtype. The largest group comprised patients with H+/M+ (51%), followed by M+ (31%), with the H+ group being the smallest at 18%. Females comprised over 70% of each group. Age, weight, BMI and PAL were comparable among SIBO groups. However, there were statistically significant differences in height between the H+/M+ and H+ groups. Patients dominant in H+/M+ exhibited a notably significantly lower height in comparison to the H+ group (*p* = 0.023). The clinical symptoms of the patients are described in our previous article [38].

### 2.2. Hydrogen and Methane Production

The detailed results for gas concentrations expressed as the area under the curve (AUC) in each group were extensively discussed in our previous article [38]. A summary of these results is presented in Figure 1. There were notable distinctions in the exhaled gases detected during the LHMBT between SIBO subtypes (*p* = 0.001).

### 2.3. Biochemical Parameters

The serum levels of vitamin D, B12, folic acid, ferritin and iron are described in Table 2. No discernible disparities were observed between the groups concerning serum vitamin D levels. Regarding levels of vitamin B12 and serum folic acid, the values remained consistent across the groups. Significant disparities in serum ferritin levels were observed between two groups, with H+ patients demonstrating elevated ferritin levels compared to H+/M+ individuals (*p* = 0.042). The median concentration of iron in each group was within the normal range, and no significant differences were observed between the groups.

Table 3 evaluates the percentage of individuals within each group exhibiting deficient, excessive or normal concentrations of the aforementioned biochemical parameters. The median concentration of vitamin D within each group was below the recommended optimal threshold. Less than half of the patients in each group, ranging from 23% to 44%, exhibited optimal levels of vitamin D concentration. Interestingly, it is noteworthy that among H+-dominant patients, 50% exhibited a borderline level of vitamin B12. In the M+ and H+/M+ groups, the majority presented with optimal concentrations of vitamin B12. More than 80% of individuals in each group exhibited optimal concentrations of folic acid. Only a small percentage of individuals exhibited either deficient or excessive serum iron levels. Although more than 50% of M+-dominant patients displayed a tendency towards iron deficiency (<10.0–15.0 ng/mL), none of the H+-dominant patients exhibited iron deficiency. However, these discrepancies were not deemed statistically significant. Moreover, over 50% of the H+ group exhibited optimal ferritin levels.

### 2.4. Energy and Macronutrients Intake

Table 4 presents the median daily intake of energy and macronutrients. No differences were observed in energy intake, BEE or TEE between the SIBO subtypes. The carbohydrate and fat intake throughout the day remained at a similar level across all three groups. The daily protein intake, both in grams and as a percentage, was similar across all groups. The fiber intake varied significantly between groups, with the M+ group exhibiting higher fiber consumption than the H+ group. Meanwhile, the H+/M+ group demonstrated similarities to both of the other groups (*p* = 0.036). Looking specifically at the consumption of fermentable sugars, only the intake of lactose differed significantly between patients. A markedly lower intake was observed in the H+ group compared to the H+/M+ group (*p* = 0.001). The consumption of SFA, MUFA and omega-3 fatty acids did not differ significantly between the groups. The ratio of omega-3/6 fatty acids was higher than what is typically recommended in the literature in all participants [39,40].

Table 5 shows the percentage of individuals who met the nutrient requirements. Only 50% of participants in the H+-dominant group met their TEE. Moreover, over 50% of patients in the M+-group exhibited a daily energy intake lower than the recommended total energy requirement for the day. However, statistical significance was not attained in this regard. Furthermore, over 40% of all participants in each group did not meet the estimated average requirement (EAR) for carbohydrate intake. All individuals in the H+ group (100%) had lower fiber intake than the AI. Only 20% of individuals in the H+-dominant group and M+ group met their fiber requirements. Moreover, it was observed that the protein intake per gram per kilogram per day exceeded the recommended norm in over 70% of all patients. Although fat intake did not differ between the groups, when considering the percentage contribution of fat, the H+ group exhibited a significantly lower fat content compared to the M+ and H+/M+ groups (*p* = 0.019). Interestingly, over 70% of patients in the M+ and H+/M+ groups had a statistically higher daily fat intake compared to the RI (*p* = 0.032). Over 50% of individuals in each group had a higher SFA intake than the recommended level.

### 2.5. Micronutrients Intake in SIBO Subtypes

Table 6 delineates the median daily consumption of selected vitamins and minerals. The intake of the examined vitamins (A, D, E, B12, folate) and minerals (iron, calcium) did not exhibit statistically significant differences between the groups.

Table 7 presents the percentage of individuals within each SIBO type who met their requirements according to micronutrients intake. An overwhelming majority of individuals in each group met the recommended daily intake for vitamin A according to the EAR. In over 87% of all patients in the study, a lower vitamin D intake was observed compared to the AI. The majority of individuals (58%) who did not meet their vitamin B12 requirements were those with H+ SIBO (*p* = 0.039) compared to M+ and H+/M+. Over 67% of all patients had an inadequate intake of calcium compared to the EAR. Less than 50% of all patients met the EAR for folate. More than 50% of all patients met the EAR for iron and RI for vitamin E.

### 2.6. Correlations

#### 2.6.1. Biochemical Parameters Correlations

Table 8 outlines the relationships between the type of gas produced by the microorganism and serum levels of biochemical parameters. An inverse correlation was observed in the H+/M+ group between serum vitamin D and the total AUC of H_2_ gas (*p* < 0.001, r = −0.6585). It was noted that higher concentrations of exhaled H_2_ were associated with significantly lower levels of serum vitamin D (Figure 2A). A similar inverse correlation was found in the same H+/M+ group regarding ferritin levels. Lower serum ferritin concentrations were associated with higher levels of H_2_ gas production (*p* = 0.001, r = −0.5648) (Figure 2B). In reference to the correlation between folate concentration and methane levels in the M+ group, it was observed that higher folate levels were solely associated with increased CH_4_ concentrations (*p* = 0.002, r = 0.6367) (Figure 2C). No correlation was detected between the levels of exhaled H_2_ or CH_4_ and serum concentrations of iron or B12 in any of the groups. Furthermore, in the H+ group, no correlations were observed in terms of biochemical parameters.

#### 2.6.2. Dietary Correlations

Table 9 summarizes the relationships between the type of gas produced by the microbiota and nutrient intake. Regarding the intake of macronutrients and selected vitamins and minerals, four correlations were identified. In the H+/M+ group (*p* = 0.001, r = −0.6462) and M+ group (*p* = 0.001, r = −0.6969), an association was observed between fiber intake and exhaled CH_4_ gas. The higher the concentration of CH_4_ gas production, the lower the intake of fiber in both groups (Figure 3A,B). Lactose intake correlated with both H_2_ and CH_4_gas production. In the H+ group, higher levels of exhaled H_2_ gas were associated with lower lactose intake in the diet (*p* = 0.027, r = −0.6338) (Figure 3C). Conversely, in the M+ group, a lower lactose intake was correlated with higher levels of exhaled CH_4_ gas *(p* = 0.002, r = −0.6444) (Figure 3D).

The remaining parameters, including energy intake, protein, fat and carbohydrate intake, as well as vitamins A, D, E, B12, folate, and minerals such as calcium and iron, were not correlated with exhaled H_2_ or CH_4_.

## 3. Discussion

It is well established that changes in the gut microbiota resulting from the overgrowth of certain types of microorganisms or disrupted relationships between them can predispose individuals to the development of SIBO [23]. It has been observed that patients with the diarrheal form of SIBO have a different gut microbiota composition compared to those with the constipated form [41]. While dietary choices are postulated to impact certain changes in microbiota profiles, there is still little that remains known on this topic [29,42,43].

To the best of our knowledge, this is the first scientific study to investigate the association between biochemical parameters, dietary intake (inclusive of nutrients and energy) and SIBO subtype. In this study, several inverse relationships were revealed between the H+/M+ group and serum ferritin, serum vitamin D and ingested fiber. Meanwhile, the M+ group showed correlations with serum folate, fiber intake and lactose intake. The H+ group only demonstrated associations with lactose intake.

### 3.1. Biochemical Parameters

#### 3.1.1. Vitamin D

In the proximal part of the small intestine, SIBO may instigate the deconjugation of bile acids which reduce the solubilization of dietary fat within micelles [44]. This alteration leads to the presence of unconjugated bile acids and a shift in their site of reabsorption, consequently leading to reduced absorption of fat-soluble vitamins (A, D, E) [44].

In our study, the inverse correlation noted solely within the H+/M+ group between vitamin D and H_2_ gas concentration could potentially be associated with the median amount of exhaled H_2_ gas during the lactose breath test. In the H+/M+ group, the concentration was 7155 ppm/min AUC, compared to 6160 ppm/min AUC in the H+ group and 2306 ppm/min AUC in the M+ group. However, the values only reached a statistically significant difference in the H+/M+ group and M+ group. It is important to consider that methane-producing Archaea are a hydrogen cross-feeder; hence, the lack of differences in H_2_ breath concentrations between the H+/M+ and H+ groups may be attributed to this phenomenon [15]. Although each of the patients examined was recently diagnosed with SIBO, evidently, higher H_2_ levels in exhaled breath contributed to the impairment of vitamin D absorption, and a more pronounced state of dysbiosis could potentially affect this group. This is corroborated by other studies, which suggest that in more severe cases of SIBO, vitamin D deficiencies will manifest [10]. To date, few studies have assessed vitamin D levels in patients with SIBO, and none of these studies has correlated vitamin D levels and SIBO subtype. In a study conducted by Zhang et al., significantly lower serum vitamin D levels were observed in pregnant patients with gestational diabetes and SIBO, compared to a control group without SIBO [45]. Among patients with concurrent SIBO and chronic pancreatitis or systemic sclerosis, no differences were noted in vitamin D levels compared to a group without SIBO [46,47]. There was a case report study involving a patient with a treatment-resistant sacral pressure sore, compounded by underlying SIBO, and observed malnutrition leading to low levels of various nutrients, including vitamin D, despite supplementation. Notably, the eradication of overgrowth alone resulted in an elevation of vitamin D levels [48]. One consequence of vitamin D deficiency is hypocalcemia, and osteoporosis is a recognized manifestation of untreated SIBO [49].While it is well established that many individuals have lower than optimal vitamin D levels or suffer from vitamin D deficiency regardless of SIBO, the results of this study add to the limited number of available studies stating that vitamin D absorption may be further compromised by excessive bacterial fermentation in patients with SIBO [44,45,48]. Further research on this topic is warranted, specifically to compare vitamin D levels between SIBO patients and a control group.

#### 3.1.2. Ferritin and Iron

This study revealed that the lowest median level of ferritin (30.5 ng/mL), reflecting early iron deficiency, was observed in the H+/M+ group. The M+ group also had low ferritin levels (35.0 ng/mL), indicative of early deficiency of iron, but was not statistically significant compared to other groups. The H+ group exhibited optimal ferritin levels (62.0 ng/mL). This may suggest that the overgrowth of both H_2_ bacteria and CH_4_Archaea predisposes one to impaired iron absorption, thus reducing ferritin levels. Interestingly, median iron levels did not differ between the groups. It is worth noting that iron is a parameter that fluctuates throughout the day and does not reflect actual iron stores, like ferritin does. Similar to vitamin D concentrations, the results demonstrated that only the amount of H_2_ gas produced in the H+/M+ group influences the serum ferritin concentration, resulting in lower ferritin levels and higher H_2_ gas concentrations in exhaled breath. This further supports the idea that a higher degree of bacterial fermentation from the excessive overgrowth of H_2_-producing bacteria contributes to impaired iron absorption and ultimately affects iron storage in cells.

Ferritin is the most sensitive indicator of iron reserves and a valuable biomarker for evaluating iron deficiency (ID). According to the World Health Organization (WHO), low ferritin levels are defined as <15 μg/L for adults and <12 μg/L for children. Nevertheless, in clinical practice, iron deficiency can be identified when ferritin levels fall below 30 μg/L [50]. Growing clinical research suggests that the gut microbiota significantly influences iron metabolism. Between 5% to 20% of iron is assimilated in the duodenum, while 80% of ingested iron is utilized by the intestinal microbiota, predominantly within the colon [51]. The intestinal microbiota utilizes iron as a cofactor in proteins essential for its survival. In environments with low iron levels, certain bacterial species such as *Eubacterium recitale*, *Roseburia* spp. experience a decrease in abundance, while members of the *Lactobacillaceae* family and *Enterobacteriaceae* family (including *Klebsiella pneumoniae*, *Salmonella* and *E. coli*) increase in abundance [52]. It is worth noting that SIBO is marked by the heightened colonization of both anaerobic and aerobic microorganisms within the small intestine, with a predominance of Gram-negative species. These may include, for example, *Escherichia coli*, *Klebsiella pneumoniae*, *Prevotella*, *Streptococcus gramineus*, *Methanobrevibacter smithii* and *Clostridium* spp. [11,53,54].

Often, in patients with insufficient iron stores in SIBO, oral iron supplementation is recommended [55]. However, it is important to remember that iron may increase the levels of pathogenic bacteria, while simultaneously reducing beneficial ones [56]. Iron is essential for methanogenic Archaea, as they depend on it for growth and metabolic processes [57]. Elevated levels of intestinal CH_4_ have been linked to decreased intestinal motility, bloating and constipation [57,58]. Therefore, considering iron supplementation in patients with iron deficiency and SIBO during bacterial eradication therapy seems to have negative consequences on the gut microbiota [57].

#### 3.1.3. Folic Acid

A distinct correlation was found between CH_4_-producing Achaea and a heightened synthesis of serum folic acid. Currently there are no published studies that assess the concentration of folic acid in blood serum among patients with different types of SIBO overgrowth. One previous study by Platovsky et al. found that participants with higher folate levels had a 1.75 times greater likelihood of having SIBO than those with normal folate levels (95% CI = 0.74–4.14) [59]. Other studies, such as those by Marie et al. and Tauber et al., only studied patients with systemic sclerosis and did not find any differences in serum folic acid between the SIBO group and the control group [47,60]. In contrast, Kaniel et al. showed that a lower serum folate level in SIBO patients after one-anastomosis gastric bypass surgery did not differ when compared to patients without SIBO [61].

Many authors have also postulated that SIBO is linked with normal or increased folate levels [18,62,63,64]. It is known that bacteria have the ability to synthesize other vitamins. The gut microbiota contributes significantly to the production and utilization offolate (vitamin B9) [65]. Based on an assessment of human gastrointestinal bacterial genomes, approximately 13.3% of bacteria are equipped with the capability of synthesizing folate de novo, while 39% have the potential to produce folic acid when supplied with additional para-aminobenzoic acid from either other microorganisms or food sources [66]. Altogether, the microbiome holds promising therapeutic potential for addressing vitamin B9 levels.

#### 3.1.4. Vitamin B12 (Cobalamin)

The results of this study demonstrate that the level of vitamin B12 was not associated with any type of bacterial overgrowth in patients with SIBO, regardless of SIBO subtype. Perhaps in the case of vitamin B12 levels, at such an early stage of overgrowth, our patients did not experience significant deficiencies in vitamin B12. There is only one study by Madigan et al. [67] that investigated the relationship between vitamin B12 and the SIBO phenotype, and revealed that methanogenic SIBO manifested only in an older population and was linked to a reduced occurrence of vitamin B12 deficiency. The majority of bacteria and Archaea lack the enzymatic ability to synthesize cobalamin de novo [68]. The absorption of vitamin B12 acquired from food can be hindered by gut bacteria that compete with the host for nutrients or by damage to cobalamin binding sites on the mucosa [63]. Moreover, *Methanobrevibacter smithii*, the primary methane-producing Archaea in the gut microbiota, is able to synthesize cobalamin. Consequently, an abundance of methanogens in SIBO is expected to have some influence on the availability of vitamin B12 to the host [68]. Future research should focus on assessing vitamin B12 levels and SIBO subtypes at a variety of patient age groups, to determine if changes in this parameter may be linked to time from diagnosis, degree of overgrowth or patient age.

### 3.2. Dietary Intake

#### 3.2.1. Fiber

A decreased consumption of fiber-rich products by patients in this study, alongside higher levels of CH_4_ in the breath test, may suggest an association between methanogens (i.e., Achaea) and poor tolerance of fiber products. Furthermore, the lower fiber intake may have contributed to exacerbating CH_4_ overgrowth. It is worth noting that in our previous study, patients in the M+ group exhibited a higher percentage of constipation compared to patients in the H+ group, who had a higher median consumption of fiber in the present study [38]. It is possible that due to existing constipation, patients increased their fiber intake compared to those in the H+-dominant group. In terms of diarrhea, our research has demonstrated that those with H_2_-dominant SIBO exhibit higher rates of diarrhea and the lowest fiber intake [38]. It is possible that individuals in this group were concerned about exacerbating their symptoms, leading to a reduction in fiber consumption. However, the decreased intake of fiber seen among subjects in this study could increase the risk of SIBO development by slowing down gut motility and disrupting the bacterial balance. Our findings are consistent with prior research, indicating that increases in *Methanobrevibacter*, *Prevotella*, *Collinsella*, *Dialister* or *Biophila* were strongly linked with a lower intake of fiber in overweight pregnant women [69]. A limited consumption of fiber encourages the proliferation of microorganisms that have the ability to utilize mucus glycoproteins secreted by the host [70]. This can potentially affect the integrity of the colonic mucus barrier and lead to the overgrowth and proliferation of pathogens [71]. According to other studies, patients with SIBO consumed less fiber than control groups [26,72]. In contrast to these findings, Cortez et al. did not find any difference in fiber intake among overweight patients with SIBO or not [73]. Saffouri et al. observed that a brief change in fiber intake can have a negative impact on a patient’s well-being. During a 7-day shift from a high-fiber diet (>11 g/1000 cal) to a low-fiber diet (<10 g/1000 cal), gastrointestinal symptoms developed in 80% of healthy volunteers, with SIBO developing in 2 out of 16 individuals [74].

#### 3.2.2. Lactose

This study confirms that a lower consumption of lactose-rich products is associated with higher concentrations of exhaled H_2_ and CH_4_. This could indicate a relationship between the amount of lactose consumed throughout the day and the extent of certain bacterial overgrowths. In our present research, the H+ group had the lowest amount of lactose consumption. This may be connected with the higher degree of diarrhea in the H_2_-dominant group that was demonstrated in our previous study [38]. According to Jo et al. [75] H_2_-dominant SIBO was found to be significantly more prevalent in patients with a lactase deficiency compared to healthy controls, with rates of 27.6% versus 6.7%, respectively. In another study, SIBO was recorded in 90% of elderly individuals diagnosed with lactose malabsorption (LM), compared to 20% of a control group [76]. The eradication of bacterial overgrowth corresponded with the alleviation of LM [76]. A retrospective analysis unveiled a prevalence of carbohydrate malabsorption among patients undergoing the lactulose breath test, and emphasized that SIBO may elevate the probability of lactose intolerance. However, intolerance in such cases is reversible with an appropriate SIBO treatment [77].

It is well established that SIBO contributes to malabsorption syndrome [78]. The overgrowth of microorganisms leads to the reduced availability of sugars consumed by bacteria [44]. Bacteria release toxins and metabolic byproducts, which can impair the integrity of the epithelial layer covering the intestinal villi [44]. Consequently, this diminishes the function of brushborder disaccharidases and impairs disaccharide digestion [77]. The increased bacterial fermentation of carbohydrates notably induces abdominal symptoms, especially an elevated production of H_2_ gas, which is primarily associated with IBS–diarrhea [79]. Signs of lactose intolerance commonly appear when lactase activity falls below 50% [80]. Furthermore, the majority of people with lactase non-persistence can manage small quantities of lactose (less than 12 g, roughly equivalent to one cup), particularly when consumed alongside other products in the diet or when divided throughout the day [80].

This is the first study to evaluate lactose consumption in patients with specific SIBO subtypes; however, seeing as a low lactose intake was found in both the H+ and M+ groups, the results align with the current literature that supports the contention that patients with SIBO are likely already consuming less lactose at baseline for symptom control. These results add to the growing body of literature addressing the dietary intake of lactose in patients with SIBO, regardless of subtype.

#### 3.2.3. Fat

This is the first study to evaluate fat intake based on SIBO subtype. The H_2_-dominant group consumed significantly less fat compared to the other two groups. Over 70% of patients in the M+ and H+/M+ groups had an excessive fat intake compared to the recommended dietary guidelines. Previous to these results, one study conducted on SpragueDawley rats revealed that animals fed a high-fat diet exhibited a higher abundance of *Methanobrevibacter smithii* in the duodenum, ileum and cecum compared to those fed a normal diet [81]. Only three other studies have assessed the total fat intake in patients with SIBO and without SIBO by using the hydrogen breath test or hydrogen–methane breath test [26,72,73]. None of the authors found any significant differences in fat intake between the SIBO group and the control group [26,72,73]. A study conducted in Iran on patients with uninvestigated dyspepsia demonstrated a higher intake of fats compared to the control group (38.3% vs. 37.4%) [82].

In clinical practice, it is often observed that patients with SIBO independently increase the fat intake in their diet due to symptoms experienced after consuming carbohydrates. However, it is essential to remember that certain types of fat can either positively or negatively affect the gut microbiota and colonic pH [30]. Laboratory studies have indeed shown that lipids in the duodenum can slow down small bowel movement and interfere with the clearance of intestinal gas, which leads to gas retention and bloating [83]. A systematic review has indicated that dietary fat may have a significant impact on the onset of symptoms related to functional dyspepsia [84].

The lower fat intake in the H+ group could have been influenced by a higher degree of diarrhea and poor fat tolerance. Depending on the type of overgrowth and current symptoms, the appropriate fat requirements should be determined.

#### 3.2.4. Vitamin B12

The consumption of vitamin B12 in the H+ group reflects theborderline serum blood levels, where over 50% of patients had significant concentrations and an intake at the borderline of normal, compared to the other two groups. None of the existing studies to date has evaluated the intake of vitamin B12 in different subtypes of SIBO.

#### 3.2.5. Other Nutrients Intake

The remaining dietary components such as energy intake, protein, total carbohydrates, fructose, SFA, MUFA PUFA, omega-3/omega-6 ratio, cholesterol, vitamin A and E, folic acid, calcium and iron, were not associated with any changes based on the type of SIBO. There are no previous studies that have assessed the intake of these parameters and SIBO subtypes.

## 4. Materials and Methods

This was an observational study performed during the period of September 2021 to January 2022 in Warsaw, Poland. This research serves as an extension of previous research, wherein the correlation between three subtypes of SIBO and selected anthropometric parameters was investigated [38]. In the context of this article, our objective is to demonstrate potential associations between biochemical parameters and dietary intake in patients with three subtypes of SIBO.

### 4.1. Sample

Adult participants between the ages of 18 to 65 years old presenting with existing abdominal symptoms occurring at least three times per month in the past six months were recruited from a dietary counseling and medical center. The study only included those individuals who newly had SIBO confirmed through a non-invasive hydrogen–methane breath test with lactulose substrate (LHMB) and had not yet started antibiotic therapy or the use of the often recommended elimination diet. Participants were required to adhere to a meticulous preparatory procedure for the study; failure to meet this criterion precluded their inclusion in the study. The exclusion criteria included individuals exhibiting a medical history indicative of eating disorders, a confirmed diagnosis of celiac disease, the presence of Inflammatory Bowel Disease (IBD), hypoglycemia, hematophobia and pregnant and lactating women. The research protocol received ethical approval from the Ethics Committee at the Institute of Human Nutrition in Warsaw, Poland (n 13/2021, date 7 May 2021). All patients provided written consent to participating in the study.

### 4.2. Lactulose Hydrogen–Methane Breath Test (LHMB)

Among all patients, lactulose breath tests were conducted and analyzed in accordance with the guidelines outlined in the North American Breath Testing Consensus Guideline (Table 10) [12].

The test enabled the measurement of gas concentrations (H_2_, CH_4_, CO_2_) in the collected breath samples of participants during exhalation. The initial gas samples were taken prior to the administration of a lactulose solution (10 g) diluted in water (150 mL), and subsequent post-lactose samples were collected over 180 min at every 10 or 20 min. The levels of H_2_ and CH_4_ in the exhaled samples were measured in parts per million (ppm). The examination utilized a QuinTron Breath Tracker (v 3.0, Warsaw, Poland) for the analysis.

### 4.3. Biochemical Assessment

Laboratory investigations were employed to assess the biochemical parameters for all the patients. We aimed to determine the levels of the following parameters: folic acid, vitamin D 25(OH)D total, ferritin, iron and vitamin B12. The participants’ venous blood samples were drawn for this analysis by qualified nurses in a certified ALAB diagnostics laboratory in Warsaw, Poland. The patients were instructed to discontinue all supplements (especially vitamin D, iron and B vitamins) one week prior to the examination and arrive at the laboratory in a fasted state before the blood draw. The serum concentrations of total vitamin D, folic acid, ferritin and vitamin B12 were determined, utilizing the direct chemiluminescence method [85]. In parallel, the iron concentration was assessed through the spectrophotometry methodology on an Abbott ANLINITY I analyzer [86]. The ALAB laboratory reported total serum vitamin D 25(OH)D concentrations according to the following categories: deficiency (<20 ng/mL), suboptimal (20–30 ng/mL), optimal (>30–50 ng/mL) and high (50–100 ng/mL). Normal levels of vitamin B12 were defined by the laboratory as (200–883 pg/mL); however, we categorized vitamin B12 concentrations as borderline (<200.0–300.0 pg/mL) or optimal (300.0–883.0 pg/mL), which was in accordance with other authors [87,88].The laboratory established normal folic acid levels as (3.1–20.5 ng/mL). We classified folic acid concentrations as low (<2.1–3.0 ng/mL), borderline (>3.0–4.0 ng/mL) and optimal (>4.0–20.5 ng/mL); in another article, a similar classification was applied [89]. The normal reference range for serum ferritin levels defined by the laboratory was (21.0–270 ng/mL) for males and (4.6–204.0 ng/mL) for females. Serum iron levels were reported by the laboratory as low (<37.0 μg/dL), optimal (37.0–145.0 μg/dL) and high (>145.0 μg/dL) for women and low (<59.0 μg/dL), optimal (59.0–158.0 μg/dL) and high (>158.0 μg/dL) for men. Given the broad range of serum ferritin norms, recognized as a significant indicator of iron stores, we have delineated the following ferritin ranges as iron deficiency (<10.0–15.0 ng/mL), early iron deficiency (>15.0–50.0 ng/mL) or optimal (>50.0–200.0 ng/mL) for women and (>50.0–270.0 ng/mL) for men, which are also widely accepted by other authors [90,91,92,93,94]. The serum concentrations of all biochemical parameters were expressed as both a median and a percentage of individuals exhibiting deficient, optimal or excessive concentrations compared to the recommended levels. Data regarding age, height and body mass were acquired from patients during the body composition analysis, which was extensively described in our previous article [38].

### 4.4. Dietary Assessment Tool

The dietary analysis relied on self-reported data. To evaluate dietary intake, each participant was instructed to meticulously record a 3-day food intake diary, encompassing two weekdays (Monday to Friday) and one weekend day (Saturday or Sunday) before considering any elimination diet. During this 3-day period, participants were asked to maintain their typical eating patterns. Patients were provided with a standard template to fill out, along with accompanying instructions asking for the following: a detailed composition of each meal, the quantity of each meal ingredient recorded in household measures or in grams, the type of culinary processing (i.e., cooking, frying, baking), all snacks, sweets, sweetened beverages, alcohol and additives such as sugar or cream for tea/coffee, as well as the time and location of meal consumption. Additionally, the template included an example of the expected meal descriptions. A dietitian verified the accuracy of all the completed records. Data from the 3-day food records were entered into the nutrition program “DietetykPro” (Warsaw, Poland), utilizing the database of the National Food and Nutrition Institute in Warsaw and the United States Department of Agriculture Foods Database [95]. Median daily values were derived from the information collected from the 3-day dietary records. The median intake of nutrients: protein (g, % energy, g/kg), carbohydrates (g, % energy), fiber (g), lactose (g), fructose (g), fat (g, % energy), saturated fatty acids (SFA) (g, % energy), monounsaturated fatty acids (MUFA) (g, % energy), polyunsaturated fatty acids (PUFA) (g, % energy), omega-3 (g), omega-6 (g), cholesterol (mg), vitamin A (μg retinol equivalents), vitamin E (mg α-tocopherol equivalents), vitamin D (μg), folate (μg), vitamin B12 (μg),calcium (mg)and iron (mg) were estimated and compared with the relevant nutritional standards and guidelines [96]. The median energy intake was juxtaposed with individual energy requirements, which the patients had estimated through the body composition analysis delineated in the preceding article [38]. The total energy expenditure (TEE) was calculated by combining the basal energy expenditure (BEE) derived from the body composition assessment and the physical activity index (PAL) [97]. Physical activity levels were evaluated through the utilization of the validated KomPAN^®^ questionnaire mentioned in the preceding article [38]. The ratio of omega-3/6 fatty acids was compared with the typical recommended ratio mentioned widely in the literature [40,98]. All the results were presented as a percentage of individuals with insufficient, adequate and excessive consumption compared to the estimated average requirement (EAR), adequate intake (AI) and reference intake (RI) norms for all nutrients [96].

### 4.5. Statistical Analysis

A statistical analysis was conducted using Statistica 13.0 Software. Both parametric and non-parametric tests were employed to analyze independent categorical variables. The serum concentration of iron (μg/dL) and the intake of energy (kcal), TEE (kcal), carbohydrates (%), protein (%, g/kg), fat (g, %), omega-6/3 ratio and folate (μg) were evaluated by Univariate ANOVA, with a posthoc Tukey test. The serum level of total vitamin D (ng/mL), vitamin B12 (pg/mL), folic acid (ng/mL) and ferritin (ng/mL);the intake of BEE (kcal), carbohydrates (g), fiber (g), lactose (g), fructose (g), protein (g),SFA (g, %), MUFA (g, %), omega-3, omega-6,PUFA (g, %), cholesterol, vitamins A (μg), E (μg), D (μg) and B12 (μg), calcium (mg) and iron (mg); and the median H_2_ (ppm/min) and CH_4_ (ppm/min) production values were analyzed with the use of the non-parametric Kruskal–Wallis test, verified on the basis of the Shapiro–Wilk test, with a posthoc Bonferroni test. All the analyses were deemed statistically significant with a threshold of *p* < 0.05. Pearson correlation analyses were employed to assess the correlation between gas production and biochemical parameters, and energy with dietary intake, in three SIBO subtypes (*p* ≤ 0.05, r ≥ ±0.3).

### 4.6. Outcome Measures

The identified gases were systematically analyzed and classified into three distinct subtypes of SIBO according to predefined guidelines:Hydrogen-dominant SIBO (H+): H_2_ > 20 ppm from the baseline within 90 min, CH_4_ < 10 ppm any time during the test;Methane-dominant SIBO (M+): H_2_ < 20 ppm from the baseline within 90 min, CH_4_ >10 ppm any time during the test;Hydrogen–methane-dominant SIBO (H+/M+): H_2_ > 20 ppm from the baseline within 90 min, CH_4_ > 10 ppm any time during the test.

The serum levels of the biochemical parameters from the laboratory tests, and the nutrient intake from the 3-day food record, were compared between the three SIBO subtypes.

## 5. Strengths and Limitations

This study had some limitations. Patients were specifically selected based on a new diagnosis of SIBO, to determine whether differences could be observed between groups at an early stage of diagnosis. We excluded patients for antibiotic therapy or the use of the recommended elimination diets during treatment, to not influence the assessment of patients’ conditions. However, it is important to note that some correlations may not have been observed due to the early stage of SIBO detection. The sample size was small, and each subgroup did not have the same number of participants, which could be another reason why the H+/M+ and M+ groups exhibited higher numbers of observed correlations, as they were larger groups than the H+ group. The 3-day food record and questionnaire were documented by the patients themselves, and so may be prone to recall bias, despite the author’s due diligence to clarify cases where doubtful data may have been provided. However, considering these limitations, significant attention was dedicated to the accuracy of the breath tests, as SIBO is often over-recognized. This research underscores the significant role of assessing both H_2_ and CH_4_ gases in patients with SIBO, in order to comprehensively evaluate their nutritional status and dietary habits. This study is also the first to thoroughly assess three types of bacterial overgrowth. When selecting an appropriate therapeutic approach for patients with SIBO, it is paramount to first diagnose the correct type of bacterial overgrowth. This allows for a more targeted treatment plan and addresses the underlying causes in terms of nutritional deficiencies and dietary irregularities. We recommend that future studies should also investigate how certain nutritional deficiencies, as well as the composition of the gut microbiota, compare between patients with different SIBO subtypes and a control group without SIBO. These investigations could be based not only on utilizing breath tests, but also through the evaluation of duodenal aspirates and stool microbiome composition by 16S rRNA gene sequencing. This would allow researchers to detect detailed changes in the profile of the gut microbiota. Furthermore, comparisons to a control group would allow for the further understanding of how SIBO, nutritional status and diet are correlated, especially among groups where certain biochemical deficiencies are common at baseline (i.e., low ferritin).

## 6. Conclusions

The results of this study suggest that the type of SIBO may have varying effects on the nutritional status of patients, leading to a range of deficiencies in the body. Furthermore, the consumption of certain nutrients by patients may result from a particular type of microbial overgrowth, predisposing them to certain gastrointestinal symptoms. Additionally, an inadequate or excessive intake of specific foods may predispose individuals to the development of a particular type of SIBO. The assessment of the nutritional status and dietary habits of patients with SIBO should be a key component of bacterial eradication therapy. The improved nutritional status of patients can lead to greater treatment efficacy and may also contribute to reducing the risk of SIBO recurrence. Often, recurrence arises from focusing solely on the functioning of the small intestine, neglecting the role of the large intestine as a primary contributor to gut microbiota.

## Figures and Tables

**Figure 1 ijms-25-07341-f001:**
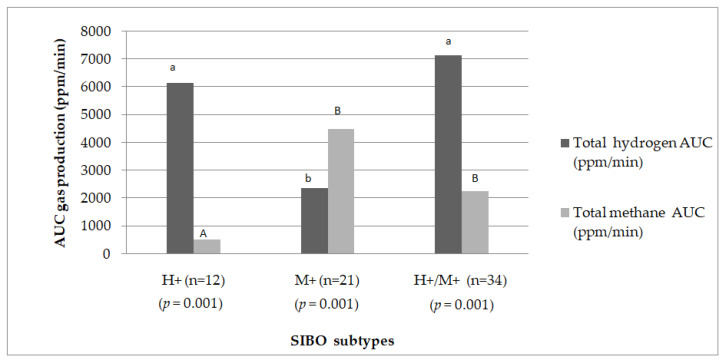
The median hydrogen and methane gas production as the area under the curve (ppm/min) in three subtypes of SIBO. *p* value is for comparison of differences between three groups, significance level of *p* = 0.05, values calculated with use of non-parametric Kruskal–Wallis test, verified on basis of Shapiro–Wilk test, a,b and A,B—Bonferroni posthoc test.

**Figure 2 ijms-25-07341-f002:**
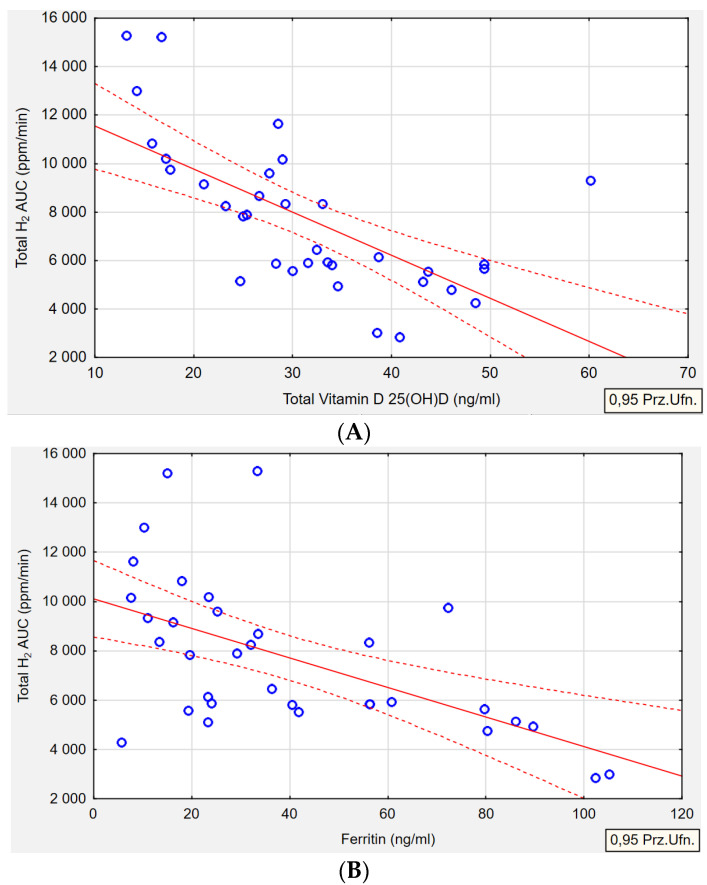
Median H_2_ concentration as area under curve (ppm/min) in H+/M+ SIBO by serum vitamin D (**A**) and serum ferritin (**B**). Median CH_4_ concentration as area under curve (ppm/min) in M+ SIBO by serum folic acid (**C**). Pearson correlation between gas production and biochemical correlation (*p* ≤ 0.05, *r* ≥ ±0.3). Red solid line—correlation line, red dotted line—confidence interval, blue circles—cases.

**Figure 3 ijms-25-07341-f003:**
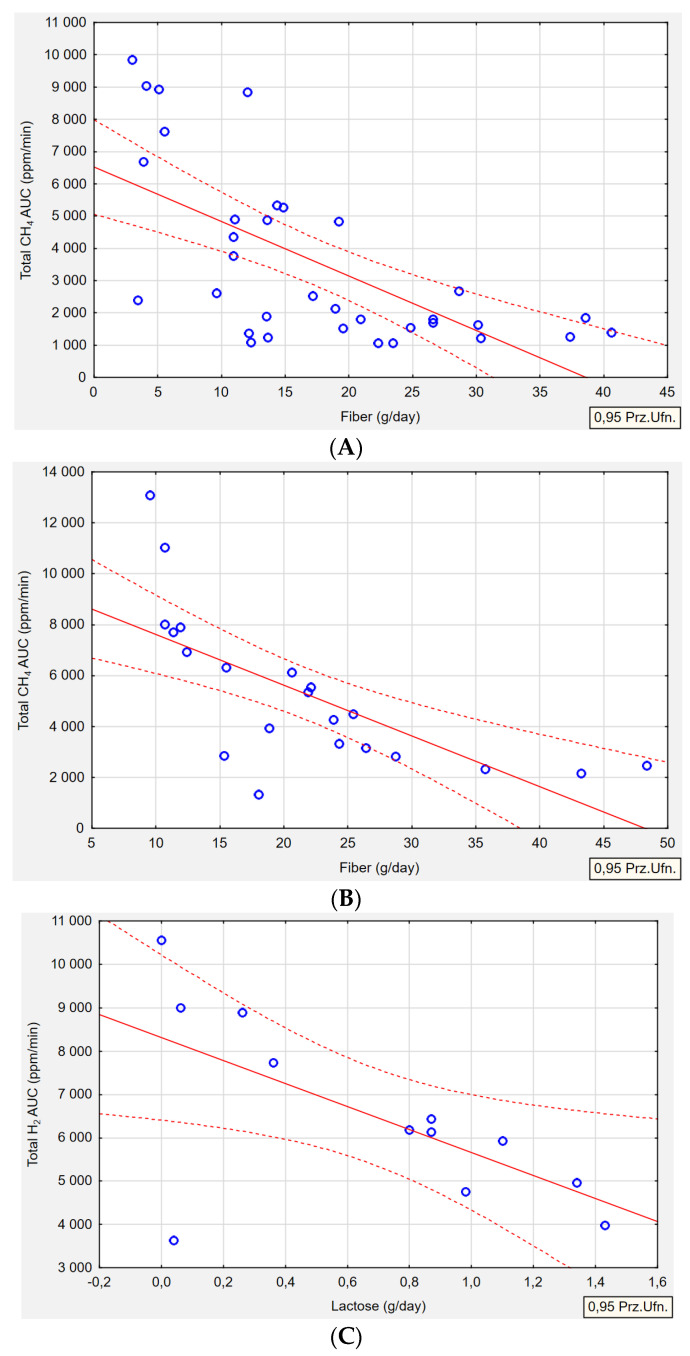
Median CH_4_ concentration as area under curve (ppm/min) in H+/M+ SIBO (**A**) and M+ SIBO (**B**) by fiber intake. Median H_2_ concentration as area under curve (ppm/min) in H+ type SIBO by lactose intake (**C**), and median CH_4_ concentration as area under curve (ppm/min) in M+ SIBO by lactose intake (**D**). Pearson correlation between gas production and nutrients intake *(p* ≤ 0.05, r ≥ ±0.3). Red solid line-correlation line, red dotted line-confidence interval, blue circles- cases.

**Table 1 ijms-25-07341-t001:** Demographic characteristics according to SIBO subtypes.

	H+ (*n* = 12)	M+ (*n* = 21)	H+/M+ (*n* = 34)	*p*-Value *
	Median (min–max)	Median (min–max)	Median (min–max)	
Age (years)	35.25 ± 11.67	33.29 ± 6.56	32.71 ± 8.23	0.776
Gender				0.921
Female	9 (75.0%)	17 (80.9)	27 (79.4%)	
Male	3 (25.0%)	4 (19.1%)	7 (20.5%)	
Height (cm)	172.5 (158.0–192.0) ^a^	170.0 (159.0–185.0) ^ab^	166.0 (153.0–189.0) ^b^	0.023 *
Weight (kg)	65.7 (45.0–109.8)	62.2 (45.8–92.3)	61.3 (39.0–86.2)	0.236
BMI (kg/m^2^)	21.8 (18.0–32.3)	21.0 (17.2–32.6)	21.7 (16.7–27.6)	0.711
PAL	1.4 (1.4–1.6)	1.4 (1.4–1.6)	1.4 (1.4–1.7)	0.487

* *p* value is for comparison of differences between three groups, significance level of *p* = 0.05, *—values calculated with use of non-parametric Kruskala–Wallisa test, verified on basis of Shapiro–Wilk test, ^a,b^—Bonferroni post hoc test.

**Table 2 ijms-25-07341-t002:** Serum levels of biochemical parameters according to SIBO subtypes.

Parameter	SIBO Subtypes			*p*-Value *
	H+ (*n* = 12)	M+ (*n* = 21)	H+/M+ (*n* = 34)	
	Median (min–max)	Median (min–max)	Median (min–max)	
Vitamin D 25(OH)D Total (ng/mL)	27.8 (13.2–37.6)	26.3 (12.4–66.0)	29.6 (13.2–60.1)	0.389
Vitamin B12 (pg/mL)	307.5 (226.0–668.0)	452.0 (218.0–846.0)	437.5 (220.8–830.0)	0.104
Folicacid (ng/mL)	6.2 (2.8–19.2)	7.1 (3.1–15.0)	6.4 (3.1–20.5)	0.707
Ferritin (ng/mL)	62.0 (16.0–104.5) ^a^	35.0 (11.0–99.0) ^ab^	30.5 (5.6–105.2) ^b^	0.042 *
Iron (μg/dL)	103.5 (84.0–168)	81.0 (35.0–157.0)	97.0 (31.0–185.0)	0.319

* *p* value is for comparison of differences between three groups, significance level of *p* = 0.05; *—values calculated with use of non-parametric Kruskala–Wallisa test, verified on basis of Shapiro-Wilk test; ^a,b^—Bonferroni posthoc test.

**Table 3 ijms-25-07341-t003:** Percentage of individuals within each SIBO subtype who met recommended serum concentrations.

Level of Parameter	SIBO Subtypes			*p*-Value *
	H+ (*n* = 12)	M+ (*n* = 21)	H+/M+ (*n* = 34)	
Vitamin D 25(OH)D Total				0.304
Deficiency (<20 ng/mL)	3 (25.0%)	6 (28.5%)	6 (17.6%)	
Suboptimal (20–30 ng/mL)	5 (41.6%)	9 (42.8%)	12 (35.3%)	
Optimal (>30–50 ng/mL)	4 (33.4%)	9 (23.8%)	15 (44.1%)	
High (50–100 ng/mL)	0 (0%)	1 (4.9%)	1 (3%)	
Vitamin B12				0.098
Borderline (<200–300 pg/mL)	6 (50%)	4 (19.0%)	7 (20.5%)	
Optimal (>300–883 pg/mL)	6 (50%)	17 (81.0%)	27 (75.5%)	
Folic acid				0.504
Low (<2.1–3.0 ng/mL)	1 (8.3%)	0 (0%)	0 (0%)	
Borderline (>3.0–4.0 ng/mL)	1 (8.3%)	1 (4.8%)	4 (11.8%)	
Optimal (>4.0–20.5 ng/mL)	10 (83.4%)	20 (95.2%)	30 (88.2%)	
Ferritin (ng/mL)				0.139
Iron deficiency (<10.0–15.0 ng/mL)	0 (0%)	11 (52.4%)	7 (20.5%)	
Early iron deficiency (>15.0–50.0 ng/mL)	5 (41.7%)	3 (14.3%)	16 (47.1%)	
Optimal (>50.0–270 ng/mL)	7 (58.3%)	7 (33.3%)	11 (32.4%)	
Iron				0.504
Low (<37.0 μg/dL W; <59.0 μg/dL M)	0 (0%)	1 (4.8%)	6 (17.6%)	
Optimal (>37.0–145.0 μg/dL W; >59.0–158.0 μg/dL M)	10 (83.4%)	19 (90.4%)	25 (73.6%)	
High (>145.0 μg/dL W; >158.0 μg/dL M)	2 (16.6%)	1 (4.8%)	3 (8.8%)	

* *p* value is for comparison of differences between three groups, significance level of *p* = 0.05.

**Table 4 ijms-25-07341-t004:** Median daily energy and macronutrients intake in SIBO subtypes.

Energy and Nutrient Intake	SIBO Subtypes			*p*-Value *
	H+ (*n* = 12)	M+ (*n* = 21)	H+/M+ (*n* = 34)	
	Median (min–max)	Median (min–max)	Median (min–max)	
Energy (kcal)	1785.5 (1330.0–2934.0)	1785.0 (1249.0–2876.0)	1695.0 (919.0–2971.0)	0.794
BEE (kcal)	1390.0 (1076.0–2122.0)	1415.0 (1186.0–1905.0)	1285.0 (1054.0–1740.0)	0.072
TEE (kcal)	1946.7 (1506.4–1506.4)	1981.0 (1660.4–2667.0)	1799.0 (1475.6–2436.0)	0.156
Carbohydrates (%E)	49.3 (34.7–70.35)	43.8 (27.3–52.6)	44.3 (12.87–73.78)	0.092
Carbohydrates (g)	241.4 (157.9–512.2)	176.6 (128.6–315.0)	214.9 (33.0–427.6)	0.274
Fiber (g)	12.3 (7.1–19.4) ^a^	20.6 (9.5–48.4) ^b^	14.6 (2.9–40.6) ^ab^	0.036 *
Fructose (g)	8.5 (1.4–24.7)	4.4 (0.0–15.1)	4.2 (0.0–16.4)	0.110
Lactose (g)	0.8 (0.0–1.4) ^a^	1.6 (0–17.3) ^ab^	5.4 (0.0–27.1) ^b^	0.001 *
Protein (%E)	15.6 (8.3–28.8)	18.2 (9.4–23.9)	17.3 (9.2–28.9)	0.860
Protein (g)	69.9 (42.4–145.3)	80.0 (38.4–171.1)	77.8 (31.4–154.7)	0.869
Protein (g/kg)	1.1 (0.6–2.5)	1.4 (0.4–2.0)	1.2 (0.5–2.3)	0.752
Fat (%E)	32.1 (17.9–50.5) ^c^	40.6 (32.2–52.3) ^d^	41.4 (26.0–64.3) ^d^	0.019
Fat (g)	62.9 (31.5–131.0)	80.6 (51.3–110.7)	86.1 (29.6–148.0)	0.210
SFA (%E)	10.2 (3.0–25.4)	13.5 (5.9–27.3)	14.2 (1.0–40.5)	0.336
SFA (g)	21.8 (9.9–66.8)	24.8 (12.5–54.3)	28.5 (1.5–71.8)	0.366
MUFA (%E)	14.9 (6.7–33.7)	16.2 (10.8–24.6)	17.9 (6.3–78.2)	0.392
MUFA (g)	33.1 (9.9–88.5)	31.5 (16.8–55.9)	37.1 (6.5–126.8)	0.673
Omega-3 (g)	1.6 (0.3–26.1)	1.7 (0.4–6.8)	1.2 (0.4–12.1)	0.878
Omega-6 (g)	7.5 (1.3–12.9)	6.8 (3.1–17.8)	7.8 (2.0–20.9)	0.693
Ratio omega-6/omega-3	4:1 (0.5:1–15.1:1)	6:1 (0.6:1–13.5:1)	7.5:1 (0.5:1–17.4:1)	0.300
PUFA (%E)	6.1 (1.5–15.5)	4.8 (3.0–12.6)	6.3 (2.4–13.5)	0.339
PUFA (g)	11.1 (4.8–40.4)	9.7 (4.3–26.4)	12.5 (2.8–34.4)	0.611
Cholesterol (mg)	296.9 (63.6–1028.0)	527.0 (117.4–1014.1)	327.7 (15.6–1159.0)	0.355

* *p* value is for comparison of differences between three groups, significance level of *p* = 0.05, *—values calculated with the use of non-parametric Kruskala–Wallisa test, verified on the basis of Shapiro–Wilk test, ^a,b^—Bonferroni posthoc test, ^c,d^—values calculated with use of Tukey parametric post-hoc test.

**Table 5 ijms-25-07341-t005:** Percentage of individuals within each SIBO subtypewho met requirements according to energy and macronutrients intake.

Level of Nutrient Intake	SIBO Subtypes			References Norm	*p*-Value *
	H+ (*n* = 12)	M+ (*n* = 21)	H+/M+ (*n* = 34)		
Energy (kcal/day) Insufficient Sufficient Excessive	4 (33.3%) 6 (50%) 2 (16.7%)	11 (52.4%) 4 (19.1%) 6 (28.5%)	15 (44.1%) 5 (14.7%) 14 (41.2%)	TEE	0.225
Carbohydrates (E%) Insufficient Sufficient Excessive	5 (41.7%) 5 (41.6%) 2 (16.7%)	11 (52.4%) 10 (47.6%) -	18 (53.0%) 13 (38.2%) 3 (8.8%)	EAR (45–56%)	0.875
Fiber (g/day) Insufficient Sufficient	12 (100%) -	15 (71.4%) 6 (28.6%)	25 (73.5%) 9 (26.5%)	AI (>25 g/day)	0.123
Protein (g/kg/day) Insufficient Sufficient Excessive	2 (16.7%) 1 (8.3%) 9 (75%)	2 (9.5%) 2 (9.5%) 17 (81%)	2 (5.9%) 3 (8.8%) 29 (85.3%)	EAR (0.73 g/kg/day)	0.766
Protein (E%) Insufficient Sufficient Excessive	1 (8.3%) 8 (66.7%) 3 (25%)	1 (4.8%) 16 (76.2%) 4 (19%)	2 (5.9%) 26 (76.5%) 6 (17.6%)	EAR (10–20%E)	0.943
Fat (E%) Sufficient Excessive	8 (66.7%) ^a^ 4 (33.3%) ^a^	5 (23.8%) ^b^ 16 (76.2%) ^b^	10 (29.4%) ^b^ 24 (70.6%) ^b^	RI (20–35%E)	0.032 *
SFA (E%) Sufficient Excessive	6 (50.0%) 6 (50.0%)	6 (26.6%) 15 (71.4%)	9 (26.5%) 25 (73.5%)	(<10%)	0.308

* *p* value is for comparison of differences between three groups, significance level of *p* = 0.05, *—values calculated with use of non-parametric Kruskala–CWallisa test, verified on basis of Shapiro–CWilk test, ^a,b^ values calculated with use of Tukey parametric posthoc test.

**Table 6 ijms-25-07341-t006:** Median daily energy and micronutrient intake in three SIBO subtypes.

Micronutrients Intake	SIBO Subtypes			*p*-Value *
	H+ (*n* = 12) Median (min–max)	M+ (*n* = 21) Median (min–max)	H+/M+ (*n* = 34) Median (min–max)	
Vitamin A (μg)	1087.5 (425.7–1704.0)	868.8 (234.9–4133.2)	923.1 (279.1–3887.2)	0.982
Vitamin D (μg)	2.1 (0.5–19.1)	3.1 (0.6–15.6)	2.53 (0.0–19.5)	0.203
Vitamin E (μg)	8.5 (3.8–18.3)	10.2 (2.2–22.0)	11.1 (2.78–44.43)	0.727
Vitamin B12 (μg)	1.7 (0.8–6.8)	3.1 (0.8–9.5)	2.7 (0.2–12.9)	0.206
Folate (μg)	227.6 (90.1–444.6)	284.6 (106.1–691.0)	245.8 (41.0–541.0)	0.432
Iron (mg)	8.8 (5.8–16.2)	9.8 (5.1–17.3)	10.0 (4.2–19.3)	0.615
Calcium (mg)	402.9 (105.0–856.9)	638.5 (7.5–1062.4)	467.4 (40.7–1860.0)	0.416

* *p* value is for comparison of differences between three groups, significance level of *p* = 0.05.

**Table 7 ijms-25-07341-t007:** Percentage of individuals within each SIBO subtype who met requirements according to micronutrients intake.

Level of Micronutrients Intake	SIBO Subtypes			References Range	*p*-Value *
	H+ (*n* = 12)	M+ (*n* = 21)	H+/M+ (*n* = 34)		
Vitamin A (μg/day) Insufficient Sufficient	2 (16.7%) 10 (83.3%)	5 (23.8%) 16 (76.2%)	16 (17.6%) 28 (82.4%)	EAR (500–630 μg)	0.827
Vitamin D (μg/day) Insufficient Sufficient	11 (91.7%) 1 (8.3%)	18 (85.7%) 3 (14.3%)	33 (97.1%) 1 (2.9%)	AI (15 μg)	0.301
Vitamin E (μg/day) Insufficient Sufficient	6 (50.0%) 6 (50.0%)	8 (38.0%) 13 (62.0%)	9 (26.5%) 25 (73.5%)	AI (8–10 μg)	0.311
Vitamin B12 (μg/day) Insufficient Sufficient	7 (58.3%) ^a^ 5 (41.7%) ^a^	3 (19.0%) ^b^ 17 (81.0%) ^b^	8 (23.5%) ^b^ 26 (76.5%) ^b^	EAR (2.0 μg)	0.039 *
Folate (μg/day) Insufficient Sufficient	9 (75.0%) 3 (25.0%)	11 (52.4%) 10 (47.6%)	24 (70.6%) 10 (29.4%)	EAR (320 μg)	0.296
Iron (mg/day) Insufficient Sufficient	6 (50.0%) 6 (50.0%)	7 (33.3%) 14 (66.7%)	8 (23.5%) 26 (76.5%)	EAR (6–8 mg)	0.235
Calcium (mg/day) Insufficient Sufficient	10 (83.3.%) 2 (16.7%)	14 (66.7%) 7 (33.3%)	25 (73.5%) 9 (26.5%)	EAR (800–1000 mg)	0.586

* *p* value is for comparison of differences between groups, significance level of *p* = 0.05, *—values calculated with use of non-parametric Kruskala–Wallisa test, verified on basis of Shapiro–Wilk test, ^a,b^—Bonferroni posthoc test.

**Table 8 ijms-25-07341-t008:** Summary of correlations between type of gas produced and serum level of biochemical parameters.

SIBO Subtypes	Relationship (Gas and Serum Level)	*p* Value and r
H+/M+	H_2_ and vitamin D	*p* < 0.001, r = −0.6585
H+/M+	H_2_ and ferritin	*p* = 0.001, r = −0.5648
M+	CH_4_ and folic acid	*p* = 0.002, r = 0.6367

r-Pearson correlation between gas production and biochemical correlation (*p* ≤ 0.05, r ≥ ±0.3).

**Table 9 ijms-25-07341-t009:** Summary of correlations between type of gas produced and nutrient intake.

SIBO Subtypes	Relationship (Gas and Dietary Intake)	*p* Value and r Value
H+/M+	CH_4_ and fiber	(*p* =0.001, r = −0.6462)
M+	CH_4_ and fiber	(*p* =0.001, r = −0.6969)
H+	H_2_ and lactose	(*p* =0.027, r = −0.6338)
M+	CH_4_ and lactose	(*p* =0.002, r = −0.6444

r-Pearson correlation between gas production and biochemical correlation (*p* ≤ 0.05, r ≥ ±0.3).

**Table 10 ijms-25-07341-t010:** Preparing individuals for LHMB.

Time before the LHMB	Conditions to Meet before Performing Test
4 weeks	Discontinuation of antibiotics
2 weeks	Avoiding probiotics
1 week	Eliminating prokinetics
3 days	Limiting fiber intake
24–48 h	Implementing carbohydrate elimination diet
12 h	Begin fasting
Day of LHMB	Continuing fast and drinking limited quantities of water (500 mL) No physical activity or smoking

## Data Availability

The data presented in this study are available on request from the corresponding author.

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
