# Peer review of "Identification of SIBO Subtypes along with Nutritional Status and Diet as Key Elements of SIBO Therapy"

_ijms, 2024, doi:10.3390/ijms25137341_

Round 1

Reviewer 1 Report

Comments and Suggestions for Authors

SIBO is a small intestine pathology that may lead to various nutritional deficiencies.  This study analyzes the nutritional status, including vitamin D, Vitamin B12, Folic acid, Ferritin, and Iron, of the three subtypes of SIBO, H+, M+, and H+/M+. It concludes that nutritional assessment and diet are crucial in SIBO therapy. This study may provide nutrition suggestions for SIBO patients. However, this study lacks a healthy intestine control group that might help identify the nutritional deficiencies for SIBO. I have several comments.

1. With regard to the statement 'In over 87% of all patients in the study, lower vitamin D intake was observed compared to the adequate intake (AI),' it's important to note that this study lacks a healthy control group. Given that many individuals have lower than optimal levels of vitamin D intake or suffer from vitamin D deficiency, without comparing the results with a control group, it's challenging to conclusively determine nutritional status as a key element of SIBO therapy.

2. As females comprised over 70% of each group, it is known that females may be more susceptible to iron deficiency than males. Have you observed differences in nutritional status between genders?

3. Why do you compare the hydrogen AUC and methane AUC within each group in Figure 1, rather than comparing the hydrogen AUC among the three groups, and likewise for methane AUC?

4. Lines 46-48. Check the grammar.

5. In Table 1, only the factor “Height” is significant. Do you have any explanation for why it is significant?

Comments on the Quality of English Language

Moderate editing of English language required

Author Response

The file with the answers is attached

Reviewer 2 Report

Comments and Suggestions for Authors

see document attached

Comments on the Quality of English Language

Author Response

The file with the answers is attached

Round 2

Reviewer 1 Report

Comments and Suggestions for Authors

The authors have addressed most of my comments.

Author Response

Dear Reviewer,

Thank you for all your work on our manuscript“Indentification of SIBO Subtypes along with Nutritional Status and Diet as Key Elements of SIBO Therapy”. Your comments and suggestions were very usefuland helped to improve the paper considerably. 

Response 1: The entire text has been re-evaluated for language and grammatical errors, and appropriate changes have been made .

Reviewer 2 Report

Comments and Suggestions for Authors

see docuement attached

Comments on the Quality of English Language

Author Response

Dear Reviewer,

Thank you for all your work on our manuscript“Indentification of SIBO Subtypes along with Nutritional Status and Diet as Key Elements of SIBO Therapy”.Your comments and suggestions were very usefuland helped to improve the paper considerably. All your suggestions have been taken into account in therecent revision of the manuscript. You can find answers to your specific comments below.

Comments 1: The manuscript ijms-3049144is the second version of an observational study in humans on the possible correlations between biochemical parameters, nutrients intake, and SIBO subtypes. A number of suggestions from the previous revision were addressed, and the text has improved, although it still requires revision in the description of Results, tables layout, etc. There still are problems particularly with alignment in tables, which are difficult to interpret. Discussion was not changed with respect to more precise references –if available-- to actual microbiota changes in connection with the various SIBO types. In summary, the text still requires a careful revision before publication.

Response 1: 

We would like to thank you for all the comments and suggestions, which are very important to us.

We have endeavored to incorporate all suggestions. In the introduction and discussion sections, we have added key information regarding microbiota changes specific to SIBO types.

Furthermore, because our study is one of the few that comprehensively assesses nutritional status and dietary habits, we aimed to explain and analyze each result extensively in the discussion section. We believe this contributes an important part to understanding the whole topic. The literature we referenced in our discussion regarding changes in gut microbiota is limited, and we relied on available sources that are worth including in the discussion.

Comments 2: 

M.smithii -as a CH4 producer– constitutes as one type of microorganism that characterizes methane dominant and/or hydrogen/methane–dominant subtypes [13]---à M. smithii -as a CH4 producer– is one of the main microbial constituents that characterize methane-dominant and/or hydrogen/methane–dominant subtypes [13].

Response 2: 

According to the literature, higher levels of M. smithii have been correlated with higher methane levels on breath tests. However, we can not conclude that M. Smithii is the main species in the hydrogen/methane dominant subtype, as this subtype also involves hydrogen-producing bacteria. Therefore, the phrase "constitutes as one type of microorganism that characterizes" is more accurate and we would like to leave it that way.

However, we have changed this paragraph in the introduction.

Comments 3: *p value is for comparison of differences among the 3 groups.Response 3: In Table 3, the percentages of individuals in each SIBO subtype are compared based on specific concentrations obtained in blood serum. We focus exclusively on the percentage of individuals in each group who, for example, had a deficiency level of vitamin D, suboptimal concentration, optimal concentration, or excessive concentration. The numerical values, such as 6 individuals, are provided solely to indicate how many people the given percentage represents within the group.

Comments 4: (reference of what is recommended in the literature).

Response 4: Thank you for your helpful comment, It has been added.

Comments 5: Figure 1. What is actually the difference between M+ and M+/H+ groups? Is it just the difference in CH4?

Response 5: 

We would like to thank you for all the comment.

Yes, the concentrations of exhaled hydrogen till  0-90th minute, till 90-180th minute, and throughout the entire test 0-180th minutes, expressed as AUC ppm/min, were similar for the hydrogen group and the hydrogen-methane group. However, differences were observed in the concentrations of exhaled methane in blood serum, both in fasting values before the lactulose challenge and throughout the entire test 0-180th minute.

We have included a table that is detailed in our first article on the assessment of anthropometric parameters in the same study group.- Table is attached

The reference to the article is repeatedly cited in the current article.

Wielgosz-Grochowska, J.P.; Domanski, N.; DrywieÅ„, M.E. Influence of Body Composition and Specific Anthropometric Parameters on SIBO Type. Nutrients 2023, 15, 1–15, doi:10.3390/nu15184035.

Comments 6: Table 3. There are problems of alignment in Table 3 which is quite difficult to interpret. Please, improve text for clarity. Idem for Tables 5 &7.

Response 6: Thank you for your helpful comment, It has been corrected.

Comments 7: Please explain EAR

Response 7: Thank you for your helpful comment, It has been corrected.

Comments 8: growing body of literature that (…) in patients with SIBO, REFERENES?

Response 8: Thank you for this comment. We made a mistake in that sentence, as we intended to emphasize that our study will add additional value to the limited number of available studies. This has already been corrected in the text.

Comments 9: The results of this study demonstrate that the level of vitamin B12 was not associated with any type of bacterial overgrowth in patients with SIBO, regardless of SIBO subtypeàBacterial growth was not studied in this work. In fact, this is a weakness of the study, and more information should be given in the discussion about the relationship between SIBO subtypes and microbiota composition.

Response 9: 

We still agree with our previous comment.

According to one of the scientific article:

Bloor, S.R.; Schutte, R.; Hobson, A.R. Oral Iron Supplementation—Gastrointestinal Side Effects and the Impact on the Gut Microbiota. Microbiol. Res. (Pavia).2021, 12, 491–502, doi:10.3390/microbiolres12020033.

Based on the results of breath tests, we are indeed unable to quantify bacterial overgrowth in terms of CFU/ml. However, we are able to assess the magnitude of overgrowth based on the concentrations of exhaled gases, following the guidelines of the North American Consensus. Therefore, when discussing the size of overgrowth, we refer to the increase in concentrations that we can analyze and evaluate based on the obtained AUC (Area Under the Curve) values.

Moreover we have added the available information about the microbiota composition.

Comments 10: higher levels of bacterial fermentation by Archaeaà Were Archaea analysed?

Response 10: 

Thank you for your comment, based on the results of breath tests, we assessed the subtypes of SIBO. Methane gas is produced by methanogens belonging to the domain Archaea.

We have corrected this sentence.

Decreased consumption of fiber-rich products by patient’s in this study, alongside higher levels of Ch4 on the breath test, may suggest an association between methanoges (Achaea) and poor tolerance to fiber products.

Comments 11: This may, be connected …-> this may be…

Response 11: Thank you for your helpful comment, It has been corrected.

Comments 12: diet due symptoms experienced-> due to

Response 12: Thank you for your helpful comment, It has been corrected.

Comments 13: Tabel 10 - Table 10

Response 13: Thank you for your helpful comment, It has been corrected.
